# Does Nirmatrelvir/Ritonavir Influence the Immune Response against SARS-CoV-2, Independently from Rebound?

**DOI:** 10.3390/microorganisms11102607

**Published:** 2023-10-22

**Authors:** Francesca Panza, Fabio Fiorino, Gabiria Pastore, Lia Fiaschi, Mario Tumbarello, Donata Medaglini, Annalisa Ciabattini, Francesca Montagnani, Massimiliano Fabbiani

**Affiliations:** 1Department of Medical Biotechnologies, University of Siena, 53100 Siena, Italy; francesca.panza@unifi.it (F.P.); lia300790@gmail.com (L.F.); mario.tumbarello@unisi.it (M.T.); 2Infectious and Tropical Diseases Unit, Azienda Ospedaliera Universitaria Senese, 53100 Siena, Italy; massimiliano.fabbiani@gmail.com; 3Laboratory of Molecular Microbiology and Biotechnology, Department of Medical Biotechnologies, University of Siena, 53100 Siena, Italy; fiorino4@unisi.it (F.F.); gabiria.pastore@unisi.it (G.P.); donata.medaglini@unisi.it (D.M.); annalisa.ciabattini@unisi.it (A.C.); 4Department of Medicine and Surgery, LUM University “Giuseppe Degennaro”, Casamassima, 70010 Bari, Italy

**Keywords:** SARS-CoV-2, COVID-19, rebound, nirmatrelvir/ritonavir, immunological response

## Abstract

Recurrence of coronavirus disease 19 (COVID-19) symptoms and SARS-CoV-2 viral load relapse have been reported in people treated with nirmatrelvir/ritonavir (NM/r). However, little is understood about the etiology of this phenomenon. Our aim was to investigate the relation between the host’s immune response and viral rebound. We described three cases of COVID-19 rebound that occurred after treatment with nirmatrelvir/ritonavir (group A). In addition, we compared spike-specific antibody response and plasma cytokine/chemokine patterns of the rebound cases with those of (i) control patients treated with nirmatrelvir/ritonavir who did not show rebound (group B), and (ii) subjects not treated with any anti-SARS-CoV-2 drug (group C). The anti-spike antibodies and plasma cytokines/chemokines were similar in groups A and B. However, we observed a higher anti-BA.2 spike IgG response in patients without antiviral treatment (group C) [geometric mean titer 210,807, 5.1- and 8.2-fold higher compared to group A (*p* = 0.039) and group B (*p* = 0.032)]. Moreover, the patients receiving antiviral treatment (groups A-B) showed higher circulating levels of platelet-derived growth factor subunit B (PDGF-BB) and vascular endothelial growth Factors (VEGF) and lower levels of interleukin-9 (IL-9), interleukine-1 receptor antagonist (IL-1 RA), and regulated upon activation normal T cell expressed and presumably secreted chemokine (RANTES) when compared to group C. In conclusion, we observed lower anti-spike IgG levels and different cytokine patterns in nirmatrelvir/ritonavir-treated patients compared to those not treated with anti-SARS-CoV-2 drugs. This suggests that early antiviral treatment, by reducing viral load and antigen presentation, could mitigate the immune response against SARS-CoV-2. The clinical relevance of such observation should be further investigated in larger populations.

## 1. Introduction

Nirmatrelvir is a recently discovered drug that inhibits SARS-CoV-2 main proteinase (Mpro), which is involved in processing the polyprotein precursors that are essential for viral production. Since Mpro activity is critical for the virus life cycle, its inhibition blocks viral replication [1]. Ritonavir is a strong CYP3A inhibitor that enhances the bioavailability of nirmatrelvir by slowing its metabolism through CYP34A. A combination of nirmatrelvir/ritonavir (NM/r) has been shown to prevent disease progression in high-risk patients when administered early in the course of infection [2], and as a consequence, it is recommended by current guidelines [3].

Recently, several cases of COVID-19 rebound occurring after oral antiviral therapy for SARS-CoV-2 have been described worldwide [4,5,6,7]. After the first reports, the CDC issued an alert on this phenomenon and defined COVID-19 rebound as a recurrence of symptoms (clinical rebound) or a new positive viral test (virological rebound) after having tested negative [8]. The clinical picture of COVID-19 rebound has not yet been clearly characterized, and it is not fully known how much it can impact the development of severe disease [9]. Most studies on COVID-19 rebound after NM/r treatment report mild symptom relapse 2–8 days after initial recovery, particularly in vaccinated subjects [4,5,7,10]. Only a few authors have investigated the isolation of SARS-CoV-2 on nasopharyngeal swabs at rebound [4,11,12,13] or reported cases of transmitted infections during rebound [7]. When complete sequencing of the SARS-CoV-2 genome was performed, virological rebound was not associated with SARS-CoV-2 emergent mutations or infections with new SARS-CoV-2 variants [4]. It should be emphasized that most studies analyzed small populations and described only a few cases of rebound. One of the largest cohorts reported a rebound rate of 0.8% (4 patients) out of a population of 483 NM/r-treated patients who were evaluated in a retrospective study [5]; all the patients were fully vaccinated and none died. Another large retrospective study conducted by Wang et al. [6] described 13,644 patients treated with NM/r (11,270 patients) or Molnupiravir (2374 patients). Among those who took NM/r, 609 patients (5.40%) experienced a viral load relapse, 662 (5.87%) experienced symptom rebound, and 87 (0.77%) were hospitalized within the 30-day period after the last day of NM/r. However, since this study is based on electronic health records, no clinical details on the rebound were described. It must be pointed out that COVID-19 clinical or virological rebound has also been observed in patients who were not treated with antivirals [14,15,16]. As a consequence, new studies are needed to better understand the relevance and the characteristics of rebound.

The reasons for rebound have not been fully elucidated, and several variables have been postulated to induce this phenomenon. Particularly, some predisposing conditions (e.g., hematological diseases), a shorter duration of antiviral therapy, a later treatment onset, and the development of drug resistance have been investigated [4,6,12,17]. Immunological factors could also be involved in the genesis of rebound; however, this aspect has been poorly explored [4,11].

Epling et al. recently performed a detailed virologic and immunologic evaluation of a group of patients with COVID-19 rebound versus those without rebound [11]. They did not find a negative effect of NM/r treatment on the immune response to SARS-CoV-2, and they observed a robust immune response after rebound, thus concluding that the risk of disease progression should be low in this circumstance. However, their findings were observed in a small subset of patients and should be confirmed in other cohorts.

Here, we analyzed COVID-19 rebound after antiviral treatment with NM/r during an epidemiological period characterized by a predominantly Omicron variant circulation, and we compared spike-specific antibodies and circulating cytokine responses in patients with and without rebound after initial recovery.

## 2. Materials and Methods

### 2.1. Study Design

We performed a retrospective observational study, enrolling three groups of SARS-CoV-2 infected patients (groups A, B and C) who were observed at our outpatient clinic in the same epidemiological period (December 2021–July 2022) that was characterized by predominant Omicron variant circulation. We only included subjects whose blood samples could be obtained close to the infection/rebound (see paragraph 2.2 for timing of blood collection).

Group A included the subjects who fulfilled the criteria for COVID-19 rebound after NM/r treatment. According to the CDC, COVID-19 rebound is defined as recurrence of symptoms (clinical rebound) associated with a new positive antigenic or molecular test for SARS-CoV-2 upon nasopharyngeal swab (virological rebound) within 7 days after having tested negative [8]. Group B consisted of control patients treated with NM/r but not experiencing any rebound. Group C consisted of SARS-CoV-2 infected patients who did not receive any antiviral treatment.

For all subjects, we described the clinical features in detail, including the evolution of symptoms and the dynamics of viral positivization/negativization over time, in order to characterize the clinical picture of rebound. Moreover, we also performed a comprehensive immunological analysis to investigate if a reduced immunological response could be the basis of rebound or if treatment with NM/r could influence immune response to SARS-CoV-2.

The study participants were recruited at the Infectious and Tropical Diseases Unit, Azienda Ospedaliera Universitaria Senese, Siena, Italy. The study was performed in compliance with all relevant ethical regulations and was approved by the local Ethical Committee for Clinical experimentation of Regione Toscana Area Vasta Sud Est (CEAVSE) (code.18869 IMMUNO_COV v1.0, 18 November 2020).

### 2.2. Analysis of the Immune Response

The antibody response was evaluated in plasma samples collected from each patient 30 days after their last SARS-CoV-2 negative antigenic or molecular swab. SARS-CoV-2 Wuhan (wild type) and BA.2 variant spike-specific IgG were tested via ELISA, as previously described [18]. In brief, recombinant wild-type and BA.2 SARS-CoV-2 spike S1+S2 proteins (Sino Biological, Beijing, China) were used for the coating of maxisorp microtiter plates (Nunc, Roskilde, Denmark) with 1 μg/mL of protein solution in PBS (Sigma-Aldrich, St. Louis, MO, USA), and incubated overnight at 4 °C. Once the plates had been blocked with 1 × PBS with 5% skimmed milk powder (AppliChem, Darmstadt, Germany) and 0.05% Tween 20 (Sigma-Aldrich, St. Louis, MO, USA), plasma samples, heated at 56 °C for 1 h, were added and titrated in two-fold dilution in duplicate in 1 × PBS with 3% skimmed milk powder and 0.05% Tween 20 (diluent buffer) and incubated for one hour at RT. Anti-human horseradish peroxidase (HRP)-conjugated IgG (diluted 1:6000, Southern Biotechologies, Birmingham, AL, USA) was then added for one hour, and the plates were developed with 3,3′,5,5′-Tetramethylbenzidine (TMB; Thermo Fisher Scientific, Waltham, MA, USA) substrate. The absorbance at 450 nm was measured using a Multiskan FC Microplate Photometer (Thermo Fisher Scientific, Waltham, MA, USA). WHO international positive (NIBSC 20/150) and negative (NIBSC 20/142) controls were included in each plate to verify the reproducibility of the assay and the positive threshold. Antibody titers were expressed as the reciprocal of the lowest sample dilution yielding a 2-fold increase in the optical density (OD) value compared to the background. We tested anti-spike antibodies, as they correlate with neutralizing antibody titers and are thought to confer protection against (re)infection. The measurement of anti-spike antibodies is commonly available and easy to perform, and it is widely accepted as a method to infer protection [19].

The analysis of cytokines and chemokines was evaluated for all patients in groups A and B, and for 3 patients in group C. Levels of IL-1beta, IL-1RA, IL-2, IL-4, IL-5, IL-6, IL-7, IL-8, IL-9, IL-10, IL-12p70, IL-13, IL-15, IL-17, basic FGF, eotaxin, G-CSF, GM-CSF, IFN-gamma, IP-10, MCP-1, MIP-1alpha, MIP-1beta, PDGF-BB, RANTES, TNF-alpha, and VEGF were tested in plasma samples collected 7 days after symptom resolution using a Luminex immunoassay (Bio-Rad, Hercules, CA, USA). The samples were collected and immediately stored at −80 °C until further use. The analysis was performed using a BioPlex Pro Human Cytokine 27-plex (Bio-Rad, Hercules, CA, USA) following the manufacturer’s protocol. The samples were read using a Bio-Plex 200 Reader (Bio-Rad, Hercules, CA, USA), and cytokine and chemokine concentrations were calculated based on standard curve data using a Bio-Plex Manager 6.2 and expressed as pg/mL.

### 2.3. Statistical Analysis

Descriptive statistics (number, proportion, median, interquartile range (IQR), range, 95% confidence intervals (CI)) were used to describe the patients’ characteristics. Continuous variables were compared between the groups using Mann–Whitney U tests. Spearman correlation tests were used for assessing the correlation between ELISA titers and the age of the subjects. A *p*-value ≤ 0.05 was considered significant. The analyses were performed using SPSS version 18.0 software package (SPSS Inc., Chicago, IL, USA).

## 3. Results

We enrolled three patients with COVID-19 clinical and virological rebound after NM/r treatment (group A) and described their clinical and virological features. Moreover, we compared their immunological response to that of three control patients treated with NM/r but not experiencing any rebound (group B) and 11 SARS-CoV-2-infected patients who did not receive any antiviral treatment and did not experience any rebound (group C). No differences in severity of COVID-19 were observed among the patients in groups A, B, or C, as they all had mild symptoms, and none required hospitalization. No data on viral load were available for our patients, since nasopharyngeal swabs were performed for diagnostic purposes only, and in routine clinical practice, viral load is not usually measured.

### 3.1. Characteristics of Rebound Cases (Group A)

All three patients in group A had comorbidities that increased the risk of disease progression, and they started NM/r 1–3 days from symptom onset (Table 1). In this group, their symptoms were mild, and they resolved after a range of 3–6 days (4, 4, and 6 days, respectively), with a negative swab after a range of 8–9 days (8, 9, and 9 days, respectively) (Table 2). Symptom rebound, accompanied by swab re-positivization, occurred after a range of 2–4 days from the first negative swab (2, 4, and 2 days, respectively), and the symptoms were mild in all cases. A second negative swab then occurred after a range of 2–7 days (4, 2, and 7 days, respectively). A detailed clinical description of rebound cases, as well as the characteristics of patients in groups B–C, are shown below.

#### 3.1.1. Rebound Case 1

At the end of June 2022, a 63-year-old Caucasian man (#1 in Table 1 and Table 2) reported a recurrence of COVID-19 symptoms after a complete 5-day treatment with nirmatrelvir/ritonavir (NM/r).

The patient, affected by liver cirrhosis and ventricular extrasystole, was on therapy with nebivolol and started NM/r 3 days after symptom onset, characterized by fever, rhinitis and cough. An antigenic swab was positive for SARS-CoV-2 on day 0. The patient was fully vaccinated (3 doses of mRNA-1237, last dose in November 2021) and reported no known previous SARS-CoV-2 infection. The initial symptoms resolved on day 6, with a marked improvement in general clinical condition. On day 5 (last day of NM/r treatment), the patient tested negative on the antigenic swab and therefore stopped his self-isolation. However, two days following the appearance of nasal congestion, otalgia and cough (day 7), the patient underwent a further SARS-CoV-2 antigenic swab, which was again positive. The patient therefore returned to his self-isolation. On day 12, his COVID-19 symptoms completely resolved, and the nasopharyngeal swab turned negative. During the rebound, the patient did not receive antiviral treatment; he took only non-steroidal anti-inflammatory drugs. No cases of intra-familial transmission were reported as a result of this rebound, since the patient lived only with his wife and son, both of whom were affected by COVID-19 during this period.

In Rebound Case 1, the spike-specific IgG titer 30 days from the last SARS-CoV-2 negative swab was 40,960 against the wild-type variant and 81,920 against the BA.2 variant. These values were 3.8- and 2.6-fold lower when compared to the geometric mean titer (GMT) of patients in group C (i.e., not treated with NM/r).

#### 3.1.2. Rebound Case 2

At the beginning of July 2022, a 58-year-old Caucasian woman (#2 in Table 1 and Table 2) reported a recurrence of COVID-19 symptoms after a complete 5-day treatment with NM/r. The patient, affected by ankylosing spondylitis and on therapy with Adalimumab for two months, received the first dose of NM/r on day 1, after the onset of a high fever, headache, sore throat, and the detection of a positive antigenic swab for SARS-CoV-2 (day 1). The patient was fully vaccinated with two doses of ChAdOx1-S and a booster dose of mRNA-1273 (last dose in December 2021) and had not had any previous SARS-CoV-2 infections. Her initial symptoms resolved on the third day of oral antiviral therapy, and on day 9, her SARS-CoV-2 antigenic swab turned negative. However, on day 12, she experienced the recurrence of fever and the onset of intense nasal congestion, rhinorrhea, and asthenia. Therefore, she underwent a new antigenic swab for SARS-CoV-2, which was again positive. The patient restarted her isolation. Three days later, the rebound symptoms resolved without the need for hospitalization. During the rebound period, the patient did not receive antiviral treatment; only non-steroidal anti-inflammatory drugs.

The patient’s husband and housemate, aged 59 (59, M), was infected at the same time with SARS-CoV-2 and was also on therapy with Adalimumab for two years due to a seronegative arthritis. He was also undergoing NM/r treatment but did not present any rebound symptoms. He was fully vaccinated with 2 doses of BNT162b2 and one of mRNA-1273 and had already had a SARS-CoV-2 infection in 2021, but he was not treated with any specific therapy. Upon his wife’s rebound, he underwent two control antigenic swabs for SARS-CoV-2, both with negative results. We enrolled him as a control (control group A).

In Rebound Case 2, the antibody spike-specific IgG titer at 30 days from the last SARS-CoV-2 negative swab was 20,480 against the wild-type variant and 40,960 against the BA.2 variant. These values were 7.5- and 5.1-fold lower when compared to the GMT of the patients in group C (i.e., not treated with NM/r).

#### 3.1.3. Rebound Case 3

In mid-July 2022, a 58-year-old Caucasian man (#3 in Table 1 and Table 2) reported a recurrence of COVID-19 symptoms after a complete 5-day treatment with NM/r. The patient, affected by severe hypertension and gallbladder calculosis, was on therapy with an ACE-inhibitor and ursodeoxycholic acid. He started NM/r treatment on day 1 after the onset of a sore throat, fever (maximum 38.5 °C of temperature), and arthromyalgias, mainly localized in the lumbar tract. The day before, he tested positive for COVID-19 on a SARS-CoV-2 RNA assay. The patient was fully vaccinated with three doses of the BNT162b2 vaccine, with his last on November 2021, and had no previous SARS-CoV-2 infections.

The COVID-19 symptoms resolved on day 4, and on day 9, the patient tested negative for SARS-CoV-2 on an antigenic swab. The day after, he stopped his isolation and resumed his social life. However, on day 11, the patient developed a symptom rebound that was mainly characterized by the onset of nasal congestion, a dry cough, posterior rhinorrhea, and the recurrence of fever. The same day (day 11), the patient again tested positive for SARS-CoV-2 RNA. The rebound symptoms lasted 5 days, and on day 18, he tested negative on an antigenic assay. During the rebound, the patient did not receive antiviral treatment; only non-steroidal anti-inflammatory drugs.

No cases of interpersonal contagion were reported, although the night before the viral load rebound, the patient, who already had nasal congestion, spent the night with two friends in an outdoor space without wearing a face mask and without maintaining a social distance. Both the patient’s friends were fully vaccinated, and one out of the two already had COVID-19 in February 2022.

For this patient, a full SARS-CoV-2 genome sequence was obtained from the first positive nasopharyngeal swab sample and found to belong to the Omicron BA.4.6 lineage.

In Rebound Case 3, the antibody spike-specific IgG titer at 30 days from the last SARS-CoV-2 negative swab was 10,240 against the wild-type variant and 20,480 against the BA.2 variant. These values were 15- and 10.3-fold lower when compared to the GMT of the patients in group C (i.e., not treated with NM/r).

### 3.2. Characteristics of Controls (Groups B and C)

Controls were selected considering the period of SARS-CoV-2 infection (December 2021–July 2022, an Omicron-like period). All the patients were fully vaccinated against SARS-CoV-2 (three doses). The data on the control patients, mixed for age and sex, are reported in Table 3.

Control group B patients received NM/r treatment but did not experience any rebound. This group included a 59-year-old man (#4 in Table 3) affected by a seronegative arthritis (husband of case 2), a 69-year-old woman (#5 in Table 3) affected by hyperlipidemia and with a recent diagnosis of melanoma in situ, and a 74-year-old man (#6 in Table 3) suffering from diabetes, high blood pressure, obesity, dyslipidemia, and gout disease. All the patients received NM/r treatment 1 day after the onset of symptoms. The antibody response in terms of specific IgG titer evaluated 30 days from the last SARS-CoV-2 negative swab was 10,240, 20,480, and 10,240 against the wild-type SARS-CoV-2 spike, and 20,480, 40,960, and 20,480 against the BA.2 variant spike, respectively.

Control group C patients had a SARS-CoV-2 infection between April and July 2022 but did not receive any antiviral treatment against SARS-CoV-2 during the infection (#7–17 in Table 3). This group consisted of 11 patients that were fully vaccinated against SARS-CoV-2 (three doses), aged from 29 to 71 years old (mean 43.6 years old) and mostly in good health.

### 3.3. Immunological Responses

Immunological responses, assessed in terms of spike-specific IgG and plasma cytokine/chemokines release, were assessed in the convalescent phase of COVID-19. The results of the antibody analysis are reported in Figure 1A,B. We observed a lower anti-spike-specific IgG response in the plasma of both rebound cases (group A) and control group B subjects compared to patients without the antiviral treatment (control group C). Specifically, no significant difference was detected regarding the antibody spike-specific IgG response against both the wild-type and BA.2 variant in group A (GMT = 20,480, 95% CI = 3660–114,586 and GMT = 40,960, 95% CI = 7321–229,171, respectively) versus control group B (GMT = 12,902, 95% CI = 4774–34,865 and GMT = 25,803, 95% CI = 9548–69,729, respectively) (*p* > 0.05 for all comparisons). However, the subjects in control group C had significantly higher antibody titers for both the anti-wild-type spike IgG [GMT = 153,834, 95% CI = 90,631–261,113, 7.5- and 11.9-fold higher compared to group A (*p* = 0.017) and control group B (*p* = 0.011)] and the anti-BA.2 spike IgG (GMT = 210,807, 95% CI = 120,221–369,649, 5.1- and 8.2-fold higher compared to group A (*p* = 0.039) and control group B (*p* = 0.032)]. This suggests that in subjects not treated with NM/r (group C), the immune response was more robust than in group A (NM/r-treated patients with rebound) and B (NM/r-treated patients without rebound).

In groups A and B, we also attempted to analyze spike-specific IgG antibody titers by considering the time elapsed between symptom onset and NM/r administration. All the patients except for one received NM/r on day 1 from symptom onset. The only patient who did not receive NM/r on day 1 (patient ID #1) showed the highest antibody levels (1–2 dilutions higher than the other subjects, see paragraph 3.1.1). No correlation was observed between the antibody titer and the age of the subjects included in the study (r = −0.28, *p* = 0.281 and r = −0.21, *p* = 0.426 for IgG anti wild type and BA.2 variant, respectively).

To investigate if cytokine release was significantly different in the rebound cases compared to the controls, we performed an extensive analysis of cytokine and chemokine profiles. (Figure 1C). The patients under antiviral treatment (groups A and B) showed higher circulating levels of PDGF-BB and VEGF [PDGF-BB median values 33.2 (IQR 30.2–35.8), 39.9 (IQR 27.5–55.5), and 21.9 pg/mL (IQR 11.7–43.9) in groups A, B, and C, respectively; VEGF median values 43.5 (IQR 42.2–43.9), 54 (IQR 23.2–60.3), and 10.6 pg/mL (IQR 3.6–23.2) in groups A, B, and C, respectively]. On the other hand, the subjects who did not receive NM/r had higher levels of IL-9, IL-1 RA, and RANTES [IL-9 median levels 5.7 (IQR 3.1–6), 5.1 (IQR 3.5–5.6), and 20.8 pg/mL (IQR 8.2–61) in groups A, B and C, respectively; IL-1 RA 14.4 (IQR 9.5–55), 25.2 (IQR 9.5–25.2), and 42 pg/mL (IQR 9.5–49.2) in groups A, B, and C, respectively; RANTES 21.7 (IQR 14.5–42.1), 33.1 (IQR 29.1–41.6), and 48.4 pg/mL (IQR 13.4–84) in groups A, B, and C, respectively]. As shown in Figure 1C, the concentration of all the cytokines detected in the plasma of the three groups was low, and pro-inflammatory cytokines/chemokines such as IFN-gamma, IL-2, TNF-alpha, IL-1 beta, IL-6, and IL-8 were completely absent. Even though small variations in cytokines were observed between the NM/r-treated and untreated subjects, the small samples size of the three groups did not allow us to detect statistically significant differences between the cases and controls.

## 4. Discussion

Starting from the beginning of the anti-SARS-CoV-2 vaccination campaign, we promoted two clinical studies aimed at deeply characterizing the spike-specific antibody and B cell response elicited by SARS-CoV-2 vaccination in cohorts of healthy [20,21] and fragile [22,23,24] subjects. Here, we focused our analysis on immunological responses to assess if they could be related to the rebound effect, which was observed in a group of our patients.

In our report, we described the clinical features of COVID-19 rebound and the virological timetable of swab negativization and re-positivization in a small group of patients. Interestingly, no cases of severe disease were observed, and all the patients had final symptoms resolution. This is in accordance with previous studies showing a general benign course of COVID-19 during rebound, with low rates of hospitalization [6]. However, some cases of complications have been reported in literature [9], and as a consequence, clinical monitoring should be warranted for these patients.

Unlike Epling et al. [11], we observed lower spike-specific IgG titers in patients treated with NM/r, independently from the development of COVID-19 rebound. The reasons for such observation are not clear yet. Since the patients included in the NM/r-treated and untreated groups were infected in the same period, the role of different circulating variants should be mitigated.

Indeed, we could speculate that this reduced immune response could be related to the early administration of NM/r itself. NM/r, by inhibiting viral replication, reduces the viral load in the initial phases of the infection [25], thus determining a lower antigen presentation to immune-competent cells. This might result in a reduced stimulation of the immune system and thus in a lower antibody response compared to untreated subjects, who are exposed to a higher viral load for longer periods. Although this pathogenetic assumption is so far purely speculative, our results could stimulate longitudinal studies to better investigate the effects of NM/r on the development of an effective immune response against SARS-CoV-2.

However, IgG titers might not explain the rebound of symptoms due to pharmacological treatment, since no relevant difference was observed between the NM/r-treated patients with rebound (group A) and those without rebound (group B). This observation should be confirmed in a larger population.

Regarding the cytokine/chemokine profile of the three groups, the concentrations of all the cytokines detected in the plasma samples were low. Moreover, the pro-inflammatory cytokines/chemokines, such as IFN-gamma, IL-2, TNF-alpha, IL-1beta, IL-6, and IL-8 were completely absent, suggesting that the inflammatory response elicited by SARS-CoV-2 infection was completely resolved in the convalescent phase of the disease. Higher serum levels of pro-inflammatory cytokines and chemokines have been observed in many patients with severe COVID-19 compared with individuals with mild disease [26]. In accordance with this observation, our patients showed mild symptoms during the acute or rebound phase of the disease, which is generally associated with a moderate inflammatory response. An apparent divergent profile was observed for VEGF, PDGF-BB, IL-9, IL1-RA, and RANTES between the NM/r-treated and untreated patients, suggesting a potential partial impact of this drug on the innate cytokine response. Indeed, no apparent relationship between cytokines/chemokine levels and rebound was found, since their levels were similar in the NM/r-treated patients with rebound (group A) compared to those without rebound (group B). Regardless, no deep conclusions can be drawn regarding the cytokine/chemokine profiles due to the small size of our sample group. Therefore, further studies on larger cohorts should be done to better understand these aspects.

Real-life observations do not change the conclusion that NM/r can markedly reduce hospitalization and death in high-risk patients [2]; but lead us to raise some questions about the NM/r treatment. Rebound might be considered a risk factor for the spread of infection, since SARS-CoV-2 has been isolated in cultures in some case reports [13]. Therefore, clinicians should be aware of this aspect, and we are wondering whether a prolonged self-isolation should be considered for patients on therapy with NM/r.

In addition, other aspects of NM/r treatment, such as pharmacokinetics, should be further investigated to understand if the duration of treatment with NM/r (i.e., a course longer than five days) or the timing of treatment onset could reduce the rebound rate. Indeed, a triple-blind phase 2 study in which immunocompromised patients are randomized and treated with NM/r for 5, 10, or 15 days is ongoing (NCT05438602) [27].

Based on our clinical experience, we recommend undergoing further SARS-CoV-2 tests if any symptoms of SARS-CoV-2 relapse appear after NM/r treatment. This will help identify the rebound and facilitate strict isolation during the rebound period to avoid any interpersonal contagion.

Some limitations should be recognized when interpreting the results of this study. A few recently published manuscripts reporting data on people vaccinated against SARS-CoV-2 have indicated that factors such as age and comorbidities can impact on the immune response to SARS-CoV2 vaccination [28,29] and may affect the level of spike-specific antibody response and its decline over time [30,31]. In our study, no significant effects of age on immune response were observed, possibly due to the small number of people included in each group, which may be a limitation of the present work.

## 5. Conclusions

The patients treated with NM/r in our study showed lower levels of circulating anti-spike IgG, independently from the rebound, compared to the control subjects not treated with NM/r and without rebound. This could be a consequence of the NM/r treatment, which is able to reduce viral load and thus potentially decrease the adaptative immune response after exposure to the virus. Although this observation does not seem to influence the incidence of rebound, it remains to be determined whether a lower antibody response during a COVID-19 episode may be associated with a more rapid decay and consequently a higher risk of early re-infection.

Despite the low number of subjects included in the cytokine analysis, some differences in the cytokine profile between NM/r-treated and untreated patients could be inferred. Their clinical relevance should be further investigated in larger populations. Therefore, future studies should focus on deeply characterizing the evolution of the humoral and cytokine immune response after SARS-CoV-2 infection in an adequate sample of patients treated or untreated with NM/r. It would also be advisable to match populations according to their age and comorbidities to limit further biases.

## Figures and Tables

**Figure 1 microorganisms-11-02607-f001:**
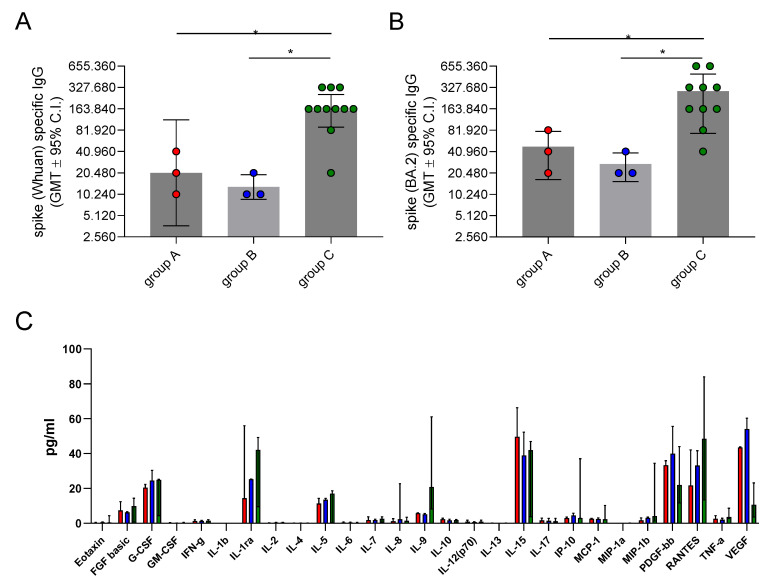
Immune response in patients with COVID-19. Spike-specific IgG titers, assessed against the Wuhan (wild type) (**A**) and BA.2 (**B**) proteins, evaluated via ELISA of plasma samples collected 30 days from the last SARS-CoV-2 negative swab in the three different groups of patients. Data are expressed as titers, calculated as the reciprocal of the lowest sample dilution yielding a 2-fold increase in the optical density (OD) value compared to the background. (**C**) Cytokine response assessed via Luminex immunoassay in plasma samples collected 7 days after symptoms resolution. Data are expressed as concentration (pg/mL). Red columns represent group A, blue columns represent group B, and green columns represent group C. Mann–Whitney U tests were used to assess the statistical differences between different groups. * *p* < 0.05.

**Table 1 microorganisms-11-02607-t001:** Clinical features of patients with COVID-19 rebound.

Patient ID	Gender, Age (y)	Comorbidities—Charlson Comorbidities Index (CCI)	Initial Symptoms	Duration of Initial Symptoms (Days)	Rebound Symptoms	Oral NM/r Start Date (Day)	Vaccine Status	COVID-19 Lineage
#1	M, 63	Cirrhosis, ventricular extrasystole—CCI 3	Fever, rhinitis and cough	6 days	Nasal congestion, otalgia and cough	Day 3	mRNA-1273, mRNA-1273, mRNA-1273	N.A
#2	F, 58	Ankylosing spondylitis—CCI 2	Fever, headache, pharyngodynia	3 days	fever, nasal congestion, rhinorrhea, asthenia	Day 1	ChAdOx1-S, ChAdOx1-S, BNT162b2	N.A
#3	M, 58	Severe hypertension, gallbladder calculosis—CCI 1	Sore throat, fever, arthromyalgias	4 days	Nasal congestion, fever, dry cough, rhinorrhea	Day 1	BNT162b2, BNT162b2, BNT162b2	Omicron BA.4.6

**Table 2 microorganisms-11-02607-t002:** Timing of the first infection and rebound.

Patient ID	1st Positive COVID Test (day), Type of Test (Antigenic/PCR)	1st Negative COVID Test (day), Type of Test (Antigenic/PCR)	Positive COVID Test at Rebound (day), Type of Test (Antigenic/PCR)	Negative COVID Test after Rebound (day), Type of Test (Antigenic/PCR)
#1	Day 3, Antigenic	Day 8, Antigenic	Day 10, Antigenic	Day 14, Antigenic
#2	Day 1, Antigenic	Day 9, Antigenic	Day 13, Antigenic	Day 15, Antigenic
#3	Day 1, PCR	Day 9, Antigenic	Day 11, PCR	Day 18, Antigenic

**Table 3 microorganisms-11-02607-t003:** Clinical data of patients without rebound (groups B and C).

Control Group	Patient ID	Gender, Age (y)	Comorbidities—Charlson Comorbidity Index (CCI)	Period of Infection	Vaccination Status and Type of Anti-SARS-CoV-2 Vaccine	Home Therapies
**B**	#4	M, 59	Seronegative arthritis—CCI 2	July 2022	BNT162b2, BNT162b2, mRNA-1273	Adalimumab
**B**	#5	F, 69	Hyperlipidemia, melanoma in situ—CCI 4	July 2022	ChAdOx1-S, ChAdOx1-S, BNT162b2	Rosuvastatin
**B**	#6	M, 74	Diabetes, high blood pressure, obesity, dyslipidemia, gout disease—CCI 4	July 2022	ChAdOx1-S, ChAdOx1-S, BNT162b2	Metformin, simvastatin, valsartan, allopurinol
**C**	#7	F, 49	None—CCI 0	May 2022	ChAdOx1-S, ChAdOx1-S, mRNA-1273	none
**C**	#8	F, 45	None—CCI 0	July 2022	BNT162b2, BNT162b2, BNT162b2	none
**C**	#9	M, 38	None—CCI 0	May 2022	BNT162b2, BNT162b2, mRNA-1273	none
**C**	#10	M, 31	Androgenetic alopecia—CCI 0	May 2022	ChAdOx1-S, ChAdOx1-S, BNT162b2	Finasteride
**C**	#11	M, 37	None—CCI 0	May 2022	ChAdOx1-S, ChAdOx1-S, mRNA-1273	none
**C**	#12	F, 71	High blood pressure, MRGE, hepatic steatosis, osteoporosis—CCI 4	April 2022	ChAdOx1-S, ChAdOx1-S, mRNA-1273	ACE inhibitor, Statin, Omeprazole. Cholecalciferol
**C**	#13	F, 67	Hyperlipidemia—CCI 2	April 2022	BNT162b2, BNT162b2, BNT162b2	Aspirin, Statin
**C**	#14	F, 29	polycystic ovary syndrome—CCI 0	April 2022	BNT162b2, BNT162b2, BNT162b2	Estroprogestinic drug
**C**	#15	F, 33	None—CCI 0	June 2022	ChAdOx1-S, ChAdOx1-S, BNT162b2	none
**C**	#16	F, 32	None—CCI 0	May 2022	INN-Ad26.COV2-S, BNT162b2	none
**C**	#17	F, 48	Celiac disease, ulcerative colitis—CCI 1	December 2021	BNT162b2, BNT162b2, mRNA-1273	none

## Data Availability

The data presented in this study are available on request from the corresponding author.

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
