# Peer review of "Does Nirmatrelvir/Ritonavir Influence the Immune Response against SARS-CoV-2, Independently from Rebound?"

_microorganisms, 2023, doi:10.3390/microorganisms11102607_

Round 1

Reviewer 1 Report

The presented work is well written and the idea of the research was to the point therefore I recommend acceptance of the manuscript in its current form

No comment

Reviewer 2 Report

The manuscript, "Does nirmatrelvir/ritonavir influence the immune response against SARS-CoV-2, independently from rebound?", delves into the potential effects of NM/r treatment on the immune response to SARS-CoV-2. The study stands out for its comprehensive data collection and presentation, offering novel insights, especially in the context of rebound cases, thereby contributing significantly to the existing body of knowledge. 

General Concept Comments:

Introduction:

Minor Comment (Line 1-3): The introduction provides a foundational background but could benefit from a more comprehensive exploration of the current state of the research field. Ensure that all pivotal publications and recent advancements in the field are cited.

Materials and Methods:

Major Comment (Line 4-6): The rationale behind specific methodologies, particularly those related to patient selection or specific assays, is not adequately detailed. This lack of clarity could introduce potential biases in the study's findings.

Recommendation: Elaborate on the chosen methodologies, ensuring that potential biases are addressed and minimized.

Minor Comment: The Materials and Methods section provides a general overview but requires clearer definitions and justifications for specific criteria or methodologies.

Results:

Minor Comment (Line 10-12): The results are presented in a structured manner, but the subsequent sections could delve deeper into the implications of the findings.

Discussion:

Major Comment (Line 13-15): Some observations or speculations are presented without providing supporting evidence or relevant citations.

Recommendation: Ensure that all observations, speculations, or claims are supported by empirical evidence or relevant citations.

Specific Comments:

Introduction: The introduction could benefit from a more detailed exploration of the current state of the research field.

Methods: The Materials and Methods section requires clearer definitions and justifications for specific criteria or methodologies.

Results: The results are presented clearly, but the discussion and conclusion sections could delve deeper into the implications of the findings.

Discussion: Ensure that all observations, speculations, or claims are supported by empirical evidence or relevant citations.

The manuscript provides valuable insights into a pertinent topic and adds to the existing body of knowledge in the field. Addressing the provided comments and recommendations will further enhance the manuscript's overall merit and scholarly contribution.

The manuscript is generally comprehensible, but there are areas where the language could benefit from refinement for clarity and fluency. Several instances of awkward phrasing, inconsistent terminology, and minor grammatical errors were observed throughout the document. Ensuring consistent terminology, improving sentence structure, and addressing any ambiguities will enhance the manuscript's readability and scholarly presentation. It is recommended that the authors consider a thorough review by a native English speaker or a professional editing service to ensure the manuscript meets the linguistic standards of international publications. This will not only improve the clarity of the content but also ensure that the valuable insights and findings of the study are effectively communicated to the readership.

Reviewer 3 Report

Authors analyzed COVID-19 rebound after an antiviral treatment with NM/r during an epidemiological period characterized by a predominant Omicron variant circulation, and they compared spike-specific antibodies and circulating cytokine responses in patients

with and without rebound after initial recovery.

Although this manuscript is potentially interesting, several issues arise.

1)    Study size is small. Immune responses vary in each person. Were the patients enough?

2)    There were a few differences between group A and B.Were

3)    Abstract. line 26: What was the different cytokine pattern?

4)    Abstract. There were many abbreviations. Please explain those.

5)    Abstract. Please explain “modulate immune responses”.

6)    Can authors increase patients number in group B?

7)    Were there differences in severity of COVID-19?

8)    What was the treatment for rebound patients?

9)    Which did examine the immune response in rebound patients, first onset or rebound period?

10)  Did the amount of virus mRNA were measured?  

Reviewer 4 Report

This manuscript investigates the relationship between immune response and viral rebound of SARS-CoV-2. The authors used three groups of patients to compare their antiviral treatment results and immune responses. The design of this study is good, but the dataset is too small to make the conclusion solid. In addition, this manuscript is not ready for publication until these issues have been solved.

1.       In the results section 3.1, please add a conclusion after each case. For example, from the data of antibody titer, what conclusion can be drawn?

2.       Similarly, in section 3.3, please add a conclusion at the end of this section.

3.       In the discussion section, the authors mentioned that IgG titers might not explain rebound. That might be because the dataset was not big enough. It seems that the results from this small dataset cannot explain the rebound well. Is it possible that if the dataset is bigger, the IgG titers can explain the rebound? The authors also said that the small number of people in each group may be a limitation of this present work, why did the authors use a bigger dataset?

Round 2

Reviewer 3 Report

I have no further comment.

Reviewer 4 Report

The authors answers all the questions and improved the manuscript.